# Use of Antiplatelet Agents Decreases the Positive Predictive Value of Fecal Immunochemical Tests for Colorectal Cancer but Does Not Affect Their Sensitivity

**DOI:** 10.3390/jpm11060497

**Published:** 2021-06-01

**Authors:** Yoonsuk Jung, Eui Im, Jinhee Lee, Hyeah Lee, Changmo Moon

**Affiliations:** 1Division of Gastroenterology, Department of Internal Medicine, Kangbuk Samsung Hospital, Sungkyunkwan University School of Medicine, Seoul 03181, Korea; ys810.jung@samsung.com; 2Division of Cardiology, Department of Internal Medicine, Yonsei University College of Medicine and Cardiovascular Center, Yongin Severance Hospital, Yongin 16995, Korea; imeui97@yuhs.ac; 3Department of Endocrinology and Metabolism, Ajou University School of Medicine, Suwon 16499, Korea; marie0715@naver.com; 4Clinical Trial Center, Ewha Womans University Mokdong Hospital, Seoul 07985, Korea; khyeah@ewha.ac.kr; 5Department of Internal Medicine, College of Medicine, Ewha Womans University, Seoul 07985, Korea; 6Inflammation-Cancer Microenvironment Research Center, Ewha Womans University, Seoul 07804, Korea

**Keywords:** fecal immunochemical test, colorectal cancer, antiplatelet agents, aspirin, clopidogrel

## Abstract

Previous studies have evaluated the effects of antithrombotic agents on the performance of fecal immunochemical tests (FITs) for the detection of colorectal cancer (CRC), but the results were inconsistent and based on small sample sizes. We studied this topic using a large-scale population-based database. Using the Korean National Cancer Screening Program Database, we compared the performance of FITs for CRC detection between users and non-users of antiplatelet agents and warfarin. Non-users were matched according to age and sex. Among 5,426,469 eligible participants, 768,733 used antiplatelet agents (mono/dual/triple therapy, n = 701,683/63,211/3839), and 19,569 used warfarin, while 4,638,167 were non-users. Among antiplatelet agents, aspirin, clopidogrel, and cilostazol ranked first, second, and third, respectively, in terms of prescription rates. Users of antiplatelet agents (3.62% vs. 4.45%; relative risk (RR): 0.83; 95% confidence interval (CI): 0.78–0.88), aspirin (3.66% vs. 4.13%; RR: 0.90; 95% CI: 0.83–0.97), and clopidogrel (3.48% vs. 4.88%; RR: 0.72; 95% CI: 0.61–0.86) had lower positive predictive values (PPVs) for CRC detection than non-users. However, there were no significant differences in PPV between cilostazol vs. non-users and warfarin users vs. non-users. For PPV, the RR (users vs. non-users) for antiplatelet monotherapy was 0.86, while the RRs for dual and triple antiplatelet therapies (excluding cilostazol) were 0.67 and 0.22, respectively. For all antithrombotic agents, the sensitivity for CRC detection was not different between users and non-users. Use of antiplatelet agents, except cilostazol, may increase the false positives without improving the sensitivity of FITs for CRC detection.

## 1. Introduction

The fecal immunochemical test (FIT) is the most commonly used non-invasive screening tool for colorectal cancer (CRC) worldwide, and it has been adopted as a population-based CRC screening program in many Western and Asian countries [1,2,3,4,5,6]. To increase the utility of the FIT, it is necessary to improve its accuracy in the detection of CRC. Additionally, to increase the cost-effectiveness of FIT-based screening programs and to decrease the number of unnecessary colonoscopies, the positive predictive value (PPV) of FITs should be high.

Antithrombotic agents, including antiplatelet agents and oral anticoagulants (OACs), are commonly used in individuals over 50 years of age, who comprise the target population for CRC screening, and the use of these agents continues to increase due to population aging [7,8]. Accordingly, the proportion of examinees who undergo FIT while taking these agents is also considerable. In theory, the use of antithrombotic agents may negatively affect the accuracy of FITs. These agents may promote bleeding from non-CRC lesions, thereby decreasing the PPV of FITs for the detection of CRC. Conversely, they may positively affect the performance of FITs. They may facilitate bleeding from early CRC lesions, resulting in increased sensitivity for CRC detection.

To date, several studies have evaluated the effect of antithrombotic agents on the performance of FITs. However, the results were inconsistent, and previous studies on this topic have had limitations and unresolved issues. Most previous studies had a small sample size and did not match age and sex between drug users and non-users, although older age and male sex are important risk factors for CRC [9,10]. Furthermore, since most of the previous studies included only FIT-positive individuals and did not include FIT-negative individuals, only the PPV could be assessed, and the sensitivity and specificity of FITs could not be compared between drug users and non-users [10,11]. A further limitation of most previous studies on antiplatelet agents is that they analyzed only aspirin use [11,12,13,14,15,16,17]. Although a few studies analyzed non-aspirin antiplatelet agents (NAAAs) or antiplatelet agents, including aspirin [11,18,19], none have precisely assessed the performance of FITs according to the type and number of antiplatelet agents used. Lastly, most previous studies only presented results for advanced colorectal neoplasia (ACRN), including both CRC and advanced adenoma, and not for CRC, which represents the more important clinical outcome.

To provide reliable results regarding the impact of antithrombotic agents on the performance of FITs for detecting CRC, large-scale studies overcoming the limitations described above are required. Therefore, we evaluated the performance (PPV, negative predictive value (NPV), sensitivity, and specificity) of FITs for the detection of CRC between users and non-users of antiplatelet agents and OACs using data from a large-scale, nationwide, population-based CRC screening program. Moreover, we assessed FIT performance according to the type of NAAA used, such as clopidogrel and cilastazol, as well as aspirin use, and assessed FIT performance according to the number of antiplatelet agents used.

## 2. Methods

### 2.1. Data Source and Study Population

The National Cancer Screening Program (NCSP) in Korea provides FITs once a year for all Koreans over 50 years of age as an initial CRC screening test and a colonoscopy free of charge as a second screening if the FIT shows positive results [20,21]. Data were extracted from the National Health Information Database (NHID) of the National Health Insurance Service (NHIS), which operates the NCSP. The study subjects consisted of participants who underwent FITs through the NCSP between 1 January 2009 and 31 December 2011.

The NHIS–NHID is encrypted and does not contain personal identifiers. This study was approved by the institutional review board of Ewha Womans University Mokdong Hospital (IRB No. 2020-11-001).

### 2.2. Study Design and Definitions of Variables

Information on age, sex, screening date, FIT results, International Classification of Diseases 10th Revision (ICD-10) codes, and drug usage were obtained from the NHIS–NHID. Patients who had both the appropriate diagnostic code (ICD-10: C18–C21, D01.0–D01.3) and the Korean national cancer registration code within 1 year after the FIT were considered to have CRC.

During the study period, the types of antiplatelet agents available in Korea were aspirin, clopidogrel, cilostazol, triflusal, limaprost indobufen, beraprost, ticlopidine, dipyridamole, and anagrelide, while warfarin was the only OAC available in Korea. Direct-acting OACs (DOACs) were not marketed during the study period. Antiplatelet agent users were defined as those who were prescribed one or more antiplatelet agents. Only patients who were prescribed these medications for at least 60 consecutive days and within 30 days of the FIT were defined as drug users [12]. Non-users were defined as those who were not prescribed any antiplatelet agents, anticoagulants, or thrombolytic agents within 2 years of the FIT.

### 2.3. Fecal Immunochemical Test

As NCSP provides a single FIT once a year, the FIT results, either qualitative or quantitative, are processed using only a single sample. We included the initial FIT result only if two or more FITs had been conducted during the study period. For the qualitative FIT test, ASAN Easy Test FOB kits (Asan Pharm, Co., Korea), SD Bioline FOB kits (SD, Co., Korea), FOB test kits (Humasis, Co., Korea), or OC-Hemocatch Light kits (Eiken Chemical, Co., Tokyo, Japan) were used. For the quantitative FIT test, OC-Sensor DIANA kits (Eiken Chemical Co.), Hemo Tech NS-1000 kits (Alfresa Pharma, Co., Osaka, Japan), or Medex HM-JACK kits (Kyowa Chemical Industry, Co., Kagawa, Japan) were used [22,23].

### 2.4. Statistical Analysis

Categorical variables among the groups were compared using the $\chi^{2}$ test, and continuous variables were compared using the Student’s *t*-test. The PPV was defined as the percentage of participants with positive FIT outcomes who were diagnosed with CRC within 1 year of the FIT, while the NPV was defined as the percentage of participants with negative FIT outcomes who were not diagnosed with CRC within 1 year of the FIT. Sensitivity was defined as the number of true positives divided by the total number of patients diagnosed with CRC within 1 year after the FIT. Specificity was defined as the number of true negatives divided by the total number of participants not diagnosed with CRC within 1 year after the FIT.

For each drug user, one non-user was randomly selected by matching for age, sex, and screening year, which may influence the performance of the FIT. These matching processes were performed independently for each drug. Modified Poisson regression with robust error variance was then used to assess the differences in the performance of the FIT between drug users and non-users [24]. In the Poisson model, as outcomes of interest, cases diagnosed with CRC within 1 year after the FIT were considered in PPV and sensitivity, and cases not diagnosed with CRC within 1 year after the FIT were considered in NPV and specificity. For PPV, for example, among those who were positive for FIT, the incidence of CRC in drug users was evaluated as a ratio compared to that in non-users. We estimated relative risk (RR) and the 95% confidence interval (CI) by adjusting for sex and age. For multiple comparison corrections on the results of each diagnostic performance indicator, we used the Benjamini–Hochberg method to control the false discovery rate at 0.05. All reported *p*-values were two-sided, and *p* < 0.05 was considered statistically significant. We calculated the power of PPV results between drug users and non-users with relatively small sample sizes. We assumed that each individual’s follow-up year is consistent and that the covariates considered in the model are independent of drug use through matching. Therefore, the power was calculated by applying the estimated RR, the size of each group, and the proportion of non-drug user groups. The sample size achieved 81% power to detect a ratio of 0.67. The test statistic used was a two-sided Z test with pooled variance, and the significance level of the test was aimed at 0.05. All data analyses were performed using the SAS software program, version 9.4 (SAS Institute, Cary, NC, USA).

## 3. Results

### 3.1. Baseline Characteristics of the Study Population

Of the 6,342,240 participants in the population, we excluded those with a previous diagnosis of CRC (n = 49,477) or inflammatory bowel disease (n = 24,064). Another 18,748 participants were excluded due to missing data (absence of data on age, sex, or screening date) and 47,578 participants were excluded due to un-certified quality assurance at the screening units. Additionally, participants who were prescribed thrombolytic agents or subcutaneous/intravenous anticoagulants within 7 days of the FIT (n = 1187) and those who received combination treatment with antiplatelet agents and anticoagulants (n = 5539) were excluded due to heterogeneity within groups. Of the remaining 6,195,647 participants, 788,302 were drug users who were prescribed antiplatelet agents or OACs (warfarin) for at least 60 consecutive days and within 30 days of the FIT, while 5,407,345 were potential drug non-users. To select only pure drug non-users specifically, we excluded a further 769,178 of these 5,407,345 participants who were prescribed antiplatelet agents, anticoagulants, or thrombolytic agents at least once within the 2 years prior to the FIT but did not meet the definition of drug user. A final total of 4,638,167 non-users and 788,302 users were included in the analysis (Figure 1).

Of the 788,302 drug users, 768,733 used antiplatelet agents and 19,569 used warfarin (Table 1). Of 768,733 antiplatelet agent users, 701,683 (91.3%) received monotherapy, while 63,211 (8.2%) received dual therapy, and 3839 (0.5%) received triple therapy. Details of the types of antiplatelet agents used in dual and triple therapies are shown in Appendix B.

Appendix A shows the demographic characteristics of the drug users and non-users in the entire study population. The proportion of men was higher in the groups of antiplatelet users (48.4%) and warfarin users (54.8%) than in the group of non-users (44.1%). The mean age was also higher in the group of antiplatelet users (64.9 ± 8.2 years) and warfarin users (65.4 ± 8.2 years) than in the group of non-users (59.6 ± 7.9 years). Table 2 shows the demographic characteristics of the 1:1 age- and sex-matched population.

### 3.2. Diagnostic Performance of FIT According to the Use of Antiplatelet Agents and Warfarin

Table 3 shows the performance of FITs for detecting CRC according to the use of antithrombotic agents in the 1:1 matched population. Among the antiplatelet agents, aspirin, clopidogrel, and cilostazol ranked first, second, and third in terms of prescription rates, respectively. Antiplatelet agent users (3.62% vs. 4.45%, RR: 0.83 (95% CI: 0.78–0.88)), aspirin users (3.66% vs. 4.13%, RR: 0.90 (95% CI: 0.83–0.97)), and clopidogrel users (3.48% vs. 4.88%, RR: 0.72 (95% CI: 0.61–0.86)) had significantly lower PPVs for CRC detection than the matched non-users. Users of a dual antiplatelet regimen that involved aspirin and clopidogrel also had lower PPVs for CRC detection than the matched non-users (3.61% vs. 5.42%, RR: 0.67 (95% CI: 0.51–0.89)). In contrast to these results, the PPVs for CRC detection were not significantly different between users and matched non-users for cilostazol (RR: 0.96 (95% CI: 0.74–1.25)), the dual antiplatelet regimen of cilostazol and aspirin (RR: 1.10 (95% CI: 0.67–1.78)), and the dual antiplatelet regimen of cilostazol and clopidogrel (RR: 1.06 (95% CI: 0.47–2.37)). With the exception of that for cilostazol, there was a tendency for a lower PPV for CRC detection in users of antiplatelet agents involved in triple therapy than in matched non-users, although the difference was not statistically significant (2.00% vs. 7.46%, RR: 0.22 (95% CI: 0.04–1.19)). Warfarin users had a similar PPV for CRC detection compared to matched non-users (3.97% vs. 4.25%, RR: 0.92 (95% CI: 0.64–1.34)).

For all antithrombotic agents studied, the sensitivities of FITs for CRC detection were not significantly different between users and matched non-users. However, the specificities for CRC detection were significantly lower in users of antiplatelet agents (total), aspirin, clopidogrel, a dual/triple antiplatelet regimen excluding cilostazol, and warfarin than in matched non-users. The NPVs were slightly higher in users of antiplatelet agents (total) and antiplatelet monotherapy.

## 4. Discussion

In this large-scale population-based study in Korea, we found that the use of antiplatelet agents was associated with a lower PPV of FITs for the detection of CRC, whereas warfarin use did not affect the PPV of FITs. When analyzed by the type of antiplatelet agents, aspirin and clopidogrel significantly lowered the PPV for CRC detection, but cilostazol did not affect the PPV. Our findings suggest that antiplatelet agents, excluding cilostazol, may increase the false-positive of FITs in the detection of CRC.

To date, only a few studies have assessed the effect of antithrombotic agents on the performance of FITs for detecting CRC. Similar to our results, a recent Norwegian study by Randel et al. [11] revealed that aspirin use was associated with a lower PPV for FIT detection of CRC (3.8% (n = 38/1008) among aspirin users vs. 6.4% (n = 65/1008) for matched non-users, *p* = 006). In contrast to the results of our study and Randel et al.’s study, three previous studies have demonstrated that the PPV of FITs for CRC detection was not affected by aspirin use. Mandelli et al. [14] showed that the PPVs of FITs for CRC detection were not significantly different between aspirin users and non-users (7.0% (n = 12/172) vs. 5.5% (n = 18/344), *p* = 0.42). Tsuji et al. [25] also reported that aspirin users had a similar PPV for CRC detection compared to non-users (4.9% (n = 4/81) vs. 5.9% (n = 55/935), adjusted OR: 0.67, 95% CI: 0.20–1.74, *p* = 0.438). Similarly, Niikura et al. [26] found no significant difference in the PPV for invasive CRC between aspirin users and non-users (3.2% (n = 13/408) vs. 3.6% (n = 15/415), *p* = 0.616). However, the small sample sizes of these three studies make it difficult to draw definite conclusions. Our study, with a vastly increased sample size (482,664 single aspirin users), provides more reliable information on the impact of aspirin use on FIT accuracy.

Our results were also consistent with those of previous studies with respect to warfarin use. In all of the previous studies [11,14,25,26], the PPVs of FITs for CRC detection were not significantly different between warfarin users and non-users. One of the reasons warfarin did not affect the PPV of FITs may be that warfarin has no anticoagulant activity within the lumen of the GI tract. The absorption rate of warfarin is more than 95%, and the intraluminal drug does not present as an active anticoagulant [27]. Although our study could not analyze the effect of DOACs on the accuracy of FITs, two previous studies have assessed this [11,26]; however, the results were inconsistent. Randel et al. [11] showed that DOAC users had a lower PPV for CRC detection than non-users (0.9% (n = 2/212) vs. 6.8% (n = 29/424), *p* = 001), whereas Niikura et al. [26] reported no significant differences in the PPV for CRC detection between DOAC users and non-users (4.1% (n = 8/197) vs. 3.1% (n = 6/196), *p* = 0.609). Further studies are needed to clarify the impact of DOAC use on the performance of FITs.

Interestingly, contrary to aspirin and clopidogrel, cilostazol did not affect the PPV for CRC detection in our study. As well, among dual antiplatelet regimens, the use of aspirin + clopidogrel was associated with a lower PPV for CRC detection, whereas the use of aspirin + cilostazol and clopidogrel + cilostazol did not reduce the PPV for CRC detection. These findings may be explained by the difference in the mechanism of action of cilostazol compared to those of other antiplatelet agents. Cilostazol decreases platelet aggregation by inhibiting phosphodiesterase-3 and increasing cyclic adenosine monophosphate (cAMP) levels in platelets [28,29]. Cilostazol also increases cAMP levels in other cells, including GI epithelial, mucus, and smooth muscle cells, which induces increased mucosal blood flow and increased secretion of mucus and HCO_3_ and inhibits muscle contraction, thereby leading to GI mucosal protection [28]. This differential mechanism of action may reduce GI bleeding or may have a protective effect against GI bleeding caused by other antiplatelet agents when used in combination with other antiplatelet agents. Indeed, some studies support this hypothesis. In a human study that compared cilostazol with aspirin, cilostazol significantly reduced the incidence of GI bleeding [29]. In another animal study, the P2Y12 receptor inhibitors, clopidogrel and ticlopidine, increased gastric bleeding and ulcerogenic responses to acidified aspirin, while cilostazol, a phosphodiesterase-3 inhibitor, suppressed these responses [28]. Based on the findings of our study and previous studies, cilostazol is unlikely to result in false positives for FITs. 

Another notable finding of our study is that as the number of antiplatelet agents increased, the PPV of the FIT further decreased. Since cilostazol did not affect the PPV for CRC detection, and the proportion of cilostazol was high in dual and triple antiplatelet therapies, we assessed the results for dual and triple antiplatelet therapies after excluding cilostazol. The RRs (users vs. non-users) for PPV with the mono, dual, and triple antiplatelet therapies were 0.86, 0.67, and 0.22, respectively. When interpreting FIT results in CRC screening, physicians need to be aware that the use of two or more antiplatelet agents can further increase the incidence of false positivity in FITs.

A recent meta-analysis of six original papers involving FIT-positive patients reported that aspirin and OACs do not affect the PPV of FITs for detecting ACRN [10]. However, a meta-analysis for CRC was not performed in this study because most previous studies on this topic only presented results for ACRN and not for CRC. Additionally, most of the studies included in this meta-analysis did not perform matching for age and sex, and antiplatelet agents were regarded as OACs because of the limited amount of available data. The heterogeneity between studies is another limitation of this meta-analysis. Considering this, it may be difficult to draw reliable conclusions from this meta-analysis regarding the PPV of FITs.

In the present study, the use of antiplatelet agents, excluding cilostazol, was associated with a lower specificity of FITs for CRC detection, while the agents did not affect the sensitivity of FITs for CRC detection. To date, only one study has evaluated the effect of aspirin on the sensitivity and specificity of FITs for CRC detection and reported no significant difference in sensitivity and specificity between aspirin users and non-users [16]. There have been several studies examining the sensitivity of FITs for ACRN detection in aspirin users. Three of these studies revealed that aspirin use did not significantly increase the sensitivity of FITs for ACRN detection [16,17,30], whereas another study demonstrated that aspirin use was associated with a markedly high sensitivity of FITs for ACRN detection [13]. It may be possible that antiplatelet agents could increase the detection of advanced adenomas by enabling bleeding in these lesions, thereby increasing the sensitivity of FIT for ACRN detection. However, given that bleeding is likely to be already present in CRC, particularly in advanced CRC, the use of antiplatelet agents is unlikely to help increase the sensitivity of FITs for CRC detection.

This is the largest study thus far to evaluate the impact of antithrombotic agents on the performance of FITs for CRC detection. Furthermore, this is the first study to assess the performance of FITs according to the type and number of antiplatelet agents. Another strength of our study is that we evaluated the sensitivity, specificity, and NPV as well as PPV. Nevertheless, it has several limitations. First, since drug use was defined using a prescription code, we could not confirm whether participants actually took the drug as prescribed. Second, because the NHIS–NHID did not contain detailed colonoscopy results, a diagnosis of CRC was confirmed by diagnostic code rather than colonoscopy. For the same reason, the effect of the use of antithrombotic drugs on the performance of FITs for detecting advanced adenoma could not be evaluated. However, since the Korean national cancer registration code was considered in the definition of CRC, it is likely that the accuracy of the CRC definition was high. Third, during the study period, various brands of FIT kits with different cutoff values were used, but detailed information such as which brand of FIT kit was used for each participant was not available in the NHIS–NHID. Accordingly, the type of FIT kit could not be adjusted for in the analyses, and it was not possible to assess whether the adjustment of the cutoff value increased the accuracy of the FIT in users of antithrombotic agents. Fourth, the subject’s FITs period was old. It was difficult to use the most recent data because it took considerable time to organize and establish the insurance data in NHID, and analysis of this data required several stages of approval and a waiting time. Fifth, we could not consider the histological types and the anatomical location of CRC because the NHIS–NHID did not contain such information. Lastly, we were unable to evaluate the effects of DOACs, which have been increasingly used in recent years.

In conclusion, the use of antiplatelet agents, including aspirin and clopidogrel, was associated with reduced PPVs of FITs for detecting CRC, while the use of cilostazol and warfarin did not affect the PPVs of FITs for CRC detection. None of the antithrombotic agents, including antiplatelet agents and warfarin, affected the sensitivity of FITs for CRC detection. Our results suggest that the use of antiplatelet agents, except cilostazol, may increase the false positives without benefiting the sensitivity of FITs for CRC detection. As the discontinuation of antithrombotic agents can increase the risk of potentially life-threatening cardiovascular events, there is no need to amend the current guidelines that recommend the continuation of antithrombotic agents prior to FITs. However, clinicians should be aware of and inform participants who are using antiplatelet agents that these agents increase the incidence of false-positive results in FITs for CRC detection.

## Figures and Tables

**Figure 1 jpm-11-00497-f001:**
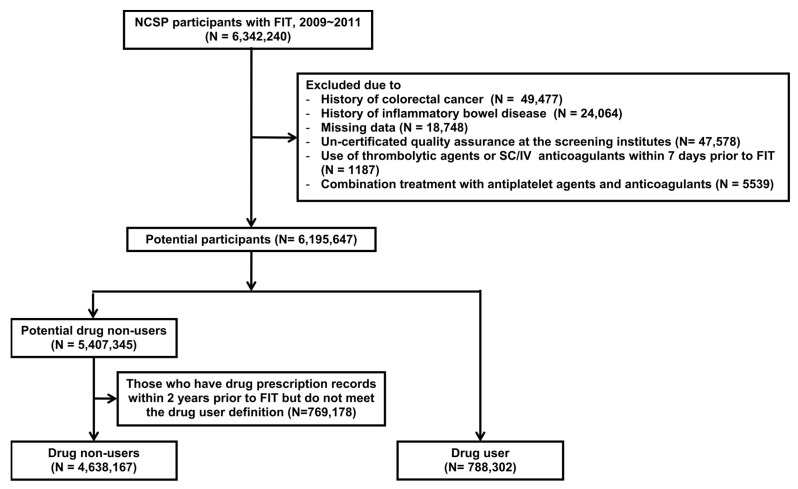
Flowchart of the study population. FIT, fecal immunochemical test; SC/IV, subcutaneous/intravenous.

**Table 1 jpm-11-00497-t001:** Number of participants using antithrombotic agents.

Drug Type	No. of Users
Antiplatelet agents	768,733
Monotherapy	701,683
Aspirin	482,664 (68.8%) *
Clopidogrel	98,468 (14.3%) *
Cilostazol	44,646 (6.4%) *
Triflusal	27,238 (3.9%) *
Limaprost	20,493 (2.9%) *
Indobufen	13,354 (1.9%) *
Beraprost	9344 (1.3%) *
Ticlopidine	4898 (0.70%) *
Dipyridamole	436 (0.06%) *
Anagrelide	142 (0.02%) *
Dual Therapy	63,211
Aspirin + Clopidogrel	31,493 (49.8%) ^†^
Aspirin + Cilostazol	10,638 (16.8%) ^†^
Other dual therapies ^‡^	21,080 (33.4%) ^†^
Triple Therapies ^‡^	3839
Anticoagulants	19,569
Warfarin	19,569

* Proportion of this agent in monotherapy; ^†^ Proportion of this agent in dual therapy; ^‡^ Details of drug types in other dual and triple therapies are shown in Appendix B.

**Table 2 jpm-11-00497-t002:** Baseline characteristics in the 1:1 matched population.

	Antiplatelet Agents, Total N = 768,720	Non-Users N = 768,720	Antiplatelet Monotherapy N = 701,674	Non-Users N = 701,674	Aspirin Monotherapy N = 482,657	Non-Users N = 482,657	Clopidogrel Monotherapy N = 98,467	Non-Users N = 98,467	Dual Antiplatelet Therapy N = 63,210	Non-Users N = 63,210	Triple Antiplatelet Therapy N = 3839	Non-Users N = 3839	Warfarin N = 19,569	Non-Users N = 19,569
Sex														
Male	372,372 (48.4)	372,372 (48.4)	332,071 (47.3)	332,071 (47.3)	227,388 (47.1)	227,388 (47.1)	50,190 (51.0)	50,190 (51.0)	37,642 (59.6)	37,642 (59.6)	2662 (69.3)	2662 (69.3)	10,715 (54.8)	10,715 (54.8)
Female	396,348 (51.6)	396,348 (51.6)	369,603 (52.7)	369,603 (52.7)	255,269 (52.9)	255,269 (52.9)	48,277 (49.0)	48,277 (49.0)	25,568 (40.5)	25,568 (40.5)	1177 (30.7)	1177 (30.7)	8854 (45.3)	8854 (45.3)
Age (years)	64.9 ± 8.2	64.9 ± 8.2	64.8 ± 8.2	64.8 ± 8.2	64.4 ± 8.2	64.4 ± 8.2	66.2 ± 8.3	66.2 ± 8.3	66.4 ± 8.0	66.4 ± 8.0	66.5 ± 7.9	66.5 ± 7.9	65.4 ± 8.2	65.4 ± 8.2

Values are presented as the number (%) or mean ± SD.

**Table 3 jpm-11-00497-t003:** Diagnostic performance of FITs for detecting CRC according to the use of antithrombotic agents in the 1:1 matched population.

Drug Types	The Number of Subjects				PPV			NPV			Sensitivity			Specificity		
	TP	FN	TN	FP	% (95% CI)	RR (95% CI)	*p* (*p_FDR_*) ^a^	% (95% CI)	RR (95% CI)	*p* (*p_FDR_*) ^a^	% (95% CI)	RR (95% CI)	*p* (*p_FDR_*) ^a^	% (95% CI)	RR (95% CI)	*p* (*p_FDR_*) ^a^
Antiplatelet agent, total																
Non-users	2111 (0.27)	1198 (0.16)	720,058 (93.67)	45,353 (5.90)	4.45 (4.26–4.63)	1.00 (ref)		99.83 (99.82–99.84)	1.00 (ref)		63.80 (62.16–65.43)	1.00 (ref)		94.07 (94.02–94.13)	1.00 (ref)	
Users	1951 (0.25)	1061 (0.14)	713,793 (92.85)	51,915 (6.75)	3.62 (3.46–3.78)	0.83 (0.78–0.88)	**<0.001** **(<0.001)**	99.85 (99.84–99.86)	1.00 (1.00–1.00)	**0.007** **(0.047)**	64.77 (63.07–66.48)	1.02 (0.98–1.05)	0.423 (0.823)	93.22 (93.16–93.28)	0.99 (0.99–0.99)	**<0.001** **(<0.001)**
Antiplatelet monotherapy																
Non-users	1821 (0.26)	1095 (0.16)	657,678 (93.73)	41,080 (5.85)	4.24 (4.05–4.44)	1.00 (ref)		99.83 (99.82–99.84)	1.00 (ref)		62.45 (60.69–64.21)	1.00 (ref)		94.12 (94.07–94.18)	1.00 (ref)	
Users	1767 (0.25)	961 (0.14)	651,777 (92.89)	47,169 (6.72)	3.61 (3.45–3.78)	0.86 (0.81–0.92)	**<0.001** **(<0.001)**	99.85 (99.84–99.86)	1.00 (1.00–1.00)	**0.006** **(0.047)**	64.77 (62.98–66.57)	1.04 (1.00–1.08)	0.069 (0.447)	93.25 (93.19–93.31)	0.99 (0.99–0.99)	**<0.001** **(<0.001)**
Aspirin monotherapy																
Non-users	1223 (0.25)	734 (0.15)	452,304 (93.71)	28,396 (5.88)	4.13 (3.90–4.36)	1.00 (ref)		99.84 (99.83–99.85)	1.00 (ref)		62.49 (60.35–64.64)	1.00 (ref)		94.09 (94.03–94.16)	1.00 (ref)	
Users	1267 (0.26)	643 (0.13)	447,352 (92.69)	33,395 (6.92)	3.66 (3.46–3.85)	0.90 (0.83–0.97)	**0.006** **(0.012)**	99.86 (99.85–99.87)	1.00 (1.00–1.00)	0.024 (0.105)	66.34 (64.22–68.45)	1.06 (1.01–1.11)	0.013 (0.164)	93.05 (92.98–93.13)	0.99 (0.99–0.99)	**<0.001** **(<0.001)**
Clopidogrel monotherapy																
Non-users	301 (0.31)	181 (0.18)	92,117 (93.55)	5868 (5.96)	4.88 (4.34–5.42)	1.00 (ref)		99.80 (99.78–99.83)	1.00 (ref)		62.45 (58.12–66.77)	1.00 (ref)		94.01 (93.86–94.16)	1.00 (ref)	
Users	232 (0.24)	151 (0.15)	91,653 (93.08)	6431 (6.53)	3.48 (3.04–3.92)	0.72 (0.61–0.86)	**<0.001** **(<0.001)**	99.84 (99.81–99.86)	1.00 (1.00–1.00)	0.107 (0.233)	60.57 (55.68–65.47)	0.97 (0.87–1.08)	0.558 (0.889)	93.44 (93.29–93.60)	0.99 (0.99–1.00)	**<0.001** **(<0.001)**
Cilostazol monotherapy																
Non-users	111 (0.25)	80 (0.18)	41,883 (93.81)	2572 (5.76)	4.14 (3.38–4.89)	1.00 (ref)		99.81 (99.77–99.85)	1.00 (ref)		58.12 (51.12–65.11)	1.00 (ref)		94.21 (94.00–94.43)	1.00 (ref)	
Users	103 (0.23)	77 (0.17)	41,932 (93.92)	2534 (5.68)	3.91 (3.17–4.65)	0.96 (0.74–1.25)	0.761 (0.888)	99.82 (99.78–99.86)	1.00 (1.00–1.00)	0.798 (0.865)	57.22 (49.99–64.45)	0.99 (0.83–1.18)	0.889 (0.889)	94.30 (94.09–94.52)	1.00 (1.00–1.00)	0.575 (0.575)
Dual antiplatelet therapy																
Non-users	221 (0.35)	92 (0.15)	58,956 (93.27)	3941 (6.23)	5.31 (4.63–5.99)	1.00 (ref)		99.84 (99.81–99.88)	1.00 (ref)		70.61 (65.56–75.65)	1.00 (ref)		93.73 (93.54–93.92)	1.00 (ref)	
Users	171 (0.27)	92 (0.15)	58,484 (92.52)	4463 (7.06)	3.69 (3.15–4.23)	0.71 (0.58–0.86)	**<0.001** **(0.002)**	99.84 (99.81–99.88)	1.00 (1.00–1.00)	0.962 (0.962)	65.02 (59.26–70.78)	0.92 (0.82–1.03)	0.147 (0.638)	92.91 (92.71–93.11)	0.99 (0.99–0.99)	**<0.001** **(<0.001)**
Aspirin + Clopidogrel																
Non-users	107 (0.34)	55 (0.17)	29,463 (93.56)	1867 (5.93)	5.42 (4.42–6.42)	1.00 (ref)		99.81 (99.76–99.86)	1.00 (ref)		66.05 (58.76–73.34)	1.00 (ref)		94.04 (93.78–94.30)	1.00 (ref)	
Users	88 (0.28)	38 (0.12)	29,016 (92.14)	2350 (7.46)	3.61 (2.87–4.35)	0.67 (0.51–0.89)	**0.005** **(0.011)**	99.87 (99.83–99.91)	1.00 (1.00–1.00)	0.090 (0.233)	69.84 (61.83–77.85)	1.06 (0.91–1.25)	0.443 (0.823)	92.51 (92.22–92.8)	0.98 (0.98–0.99)	**<0.001** **(<0.001)**
Aspirin + Cilostazol																
Non-users	29 (0.27)	16 (0.15)	9942 (93.46)	651 (6.12)	4.26 (2.75–5.78)	1.00 (ref)		99.84 (99.76–99.92)	1.00 (ref)		64.44 (50.46–78.43)	1.00 (ref)		93.85 (93.40–94.31)	1.00 (ref)	
Users	33 (0.31)	23 (0.22)	9903 (93.09)	679 (6.38)	4.63 (3.09–6.18)	1.10 (0.67–1.78)	0.713 (0.888)	99.77 (99.67–99.86)	1.00 (1.00–1.00)	0.256 (0.475)	58.93 (46.04–71.81)	0.96 (0.71–1.30)	0.806 (0.889)	93.58 (93.12–94.05)	1.00 (0.99–1.00)	0.414 (0.490)
Clopidogrel + Cilostazol																
Non-users	12 (0.31)	11 (0.28)	3609 (93.35)	234 (6.05)	4.88 (2.19–7.57)	1.00 (ref)		99.70 (99.52–99.88)	1.00 (ref)		52.17 (31.76–72.59)	1.00 (ref)		93.91 (93.15–94.67)	1.00 (ref)	
Users	12 (0.31)	7 (0.18)	3626 (93.79)	221 (5.72)	5.15 (2.31–7.99)	1.06 (0.47–2.37)	0.885 (0.888)	99.81 (99.66–99.95)	1.00 (1.00–1.00)	0.337 (0.536)	63.16 (41.47–84.85)	1.14 (0.66–1.96)	0.635 (0.889)	94.26 (93.52–94.99)	1.00 (0.99–1.02)	0.521 (0.564)
Dual antiplatelet therapy excluding cilostazol																
Non-users	160 (0.34)	76 (0.16)	43,520 (93.29)	2893 (6.20)	5.24 (4.45–6.03)	1.00 (ref)		99.83 (99.79–99.86)	1.00 (ref)		67.80 (61.84–73.76)	1.00 (ref)		93.77 (93.55–93.99)	1.00 (ref)	
Users	123 (0.26)	56 (0.12)	43,013 (92.21)	3457 (7.41)	3.44 (2.84–4.03)	0.67 (0.53–0.84)	**<0.001** **(0.002)**	99.87 (99.84–99.90)	1.00 (1.00–1.00)	0.093 (0.233)	68.72 (61.92–75.51)	1.01 (0.89–1.15)	0.867 (0.889)	92.56 (92.32–92.80)	0.99 (0.98–0.99)	**<0.001** **(<0.001)**
Triple antiplatelet therapy																
Non-users	11 (0.29)	7 (0.18)	3577 (93.18)	244 (6.36)	4.31 (1.82–6.81)	1.00 (ref)		99.80 (99.66–99.95)	1.00 (ref)		61.11 (38.59–83.63)	1.00 (ref)		93.61 (92.84–94.39)	1.00 (ref)	
Users	13 (0.34)	8 (0.21)	3535 (92.08)	283 (7.37)	4.39 (2.06–6.73)	1.06 (0.49–2.29)	0.888 (0.888)	99.77 (99.62–99.93)	1.00 (1.00–1.00)	0.785 (0.865)	61.90 (41.13–82.68)	1.07 (0.66–1.74)	0.776 (0.889)	92.59 (91.76–93.42)	0.99 (0.98–1.00)	0.077 (0.100)
Triple antiplatelet therapy excluding cilostazol																
Non-users	5 (0.52)	1 (0.10)	889 (92.89)	62 (6.48)	7.46 (1.17–13.76)	1.00 (ref)		99.89 (99.67–100.0)	1.00 (ref)		83.33 (53.51–100.0)	1.00 (ref)		93.48 (91.91–95.05)	1.00 (ref)	
Users	2 (0.21)	2 (0.21)	855 (89.34)	98 (10.24)	2.00 (0.00–4.74)	0.22 (0.04–1.19)	0.078 (0.127)	99.77 (99.44–100.0)	1.00 (0.99–1.00)	0.557 (0.724)	50.00 (1.00–99.00)	0.67 (0.28–1.61)	0.371 (0.823)	89.72 (87.79–91.65)	0.96 (0.93–0.99)	**0.003** **(0.005)**
Warfarin																
Non-users	52 (0.27)	40 (0.2)	18,305 (93.54)	1172 (5.99)	4.25 (3.12–5.38)	1.00 (ref)		99.78 (99.71–99.85)	1.00 (ref)		56.52 (46.39–66.65)	1.00 (ref)		93.98 (93.65–94.32)	1.00 (ref)	
Users	56 (0.29)	32 (0.16)	18,127 (92.63)	1354 (6.92)	3.97 (2.95–4.99)	0.92 (0.64–1.34)	0.674 (0.888)	99.82 (99.76–99.88)	1.00 (1.00–1.00)	0.371 (0.536)	63.64 (53.59–73.69)	1.13 (0.89–1.43)	0.332 (0.823)	93.05 (92.69–93.41)	0.99 (0.98–1.00)	**<0.001** **(<0.001)**

Statistically significant results are shown in bold. FIT, fecal immunochemical test; CRC, colorectal cancer; TP, true positive; FN, false negative; TN, true negative; FP, false positive; PPV, positive predictive value; NPV, negative predictive value; RR, relative risk; CI, confidence interval. ^a^ The false discovery rates (FDRs) were determined using the Benjamini and Hochberg test for multiple comparisons correction.

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
