# Peer review of "Use of Antiplatelet Agents Decreases the Positive Predictive Value of Fecal Immunochemical Tests for Colorectal Cancer but Does Not Affect Their Sensitivity"

_jpm, 2021, doi:10.3390/jpm11060497_

Round 1

Reviewer 1 Report

The scientific work seems interesting but the rationale of comparing two drugs with different biomolecular actions, namely antiplatelet agents and anticoagulants, is not recognized. Furthermore, it would be interesting to compare these data with the histological types of the tumors examined and also by dividing the anatomical location (ex: ascending, transverse, descending, rectum colon).

Author Response

Reviewer 1.

The scientific work seems interesting but the rationale of comparing two drugs with different biomolecular actions, namely antiplatelet agents and anticoagulants, is not recognized. Furthermore, it would be interesting to compare these data with the histological types of the tumors examined and also by dividing the anatomical location (ex: ascending, transverse, descending, rectum colon).

Reply: Thank you for your comments. Our study compared between antiplatelet agent users vs. non-users and between anticoagulant users vs. non-users. Our study did not compare between antiplatelet agent user and anticoagulant users. Unfortunately, the NHIS–NHID (National Health Insurance Service-National Health Information Database) does not contain information on the histological types and the anatomical location of CRC. Accordingly, we could not evaluate these factors. We added this limitation in the Discussion section as follows: “Fifth, we could not consider the histological types and the anatomical location of CRC because the NHIS–NHID did not contain such information.”

Reviewer 2 Report

  1. The data analysed in the paper was for those having FIT between 2009 and 2011. As this is over a decade ago now, please explain why more recent data was not used. Also, please comment on applicability of the results given that the data was this old, since presumably FIT will have evolved since then. 
  2. Your p-value threshold of 0.05 should be corrected to reflect the number of hypothesis tests conducted. Please ensure a correction is applied and justified within the manuscript, otherwise the conclusions could be misleading. 
  3. You specify combination treatment of antiplatelet agents and anticoagulants as an exclusion criterion. Would it not have been of interest to also investigate this group of patients ?
  4. The p-values presented in Table 2 are meaningless without any correction for multiple testing due to the large amount of comparisons undertaken. Indeed, the relevance of the information within this table is questionable given that the majority of non-drug-users were excluded from the analysis as they were removed during the matching process. My preference would be to exclude this table altogether, and just keep table 3 which relates to the actual participants studied. 
  5. Did you consider any adjustments in your analyses for type of test undertaken, given that there was large variation in the type of test. 
  6. Why did you choose Poisson regression modelling to test for difference in PPV between the two groups? In addition, what covariates were adjusted for in the regression model ? As you are comparing proportions it would be more usual to use a hypothesis test such as the chi-square test, or if wanting to adjust for covariates, a logistic regression approach. 
  7. What is the justification for undertaking an analysis of the overall dataset in addition to the matched analysis ? The former seems to make the latter redundant. I would suggest that the overall analysis is excluded, and instead the paper focusses on the matched analysis only. 
  8. What is the 'univariable' analysis referred to in Table 5 ? Is this the same as the 'crude' analysis ? Please ensure consistency of terminology throughout. Indeed, please ensure that all methods used in producing the reported results are reported in the methods section, for transparency. 
  9. In lines 316-317, on what basis are you claiming " almost all participants would have taken antithrombotic drugs on the day of as well as before the FIT. " As far as I am aware there were no measurements of extent of adherence available in the dataset and so this statement could be misleading. The authors may wish to explore known rates of adherence for the studied drugs, and comment accordingly on the potential impact of adherence on the study estimates. 
  10. Please provide a power calculation to illustrate available power for the various tests undertaken. 

Author Response

Reviewer 2.

1.The data analysed in the paper was for those having FIT between 2009 and 2011. As this is over a decade ago now, please explain why more recent data was not used. Also, please comment on applicability of the results given that the data was this old, since presumably FIT will have evolved since then. 

Reply: Thank you for your valuable comments. As the reviewer pointed out, we agree that the subject's test period was old. However, since this study was conducted by analyzing the data of the National Health Information Database (NHID) of the National Health Insurance Service (NHIS), there were limitations in using the recent data. First, since it took considerable long time to organize and establish the insurance data in NHID. Second, several stages of approval and a waiting time were required for us to use this data. In addition, we conducted statistical analyzes many times during the research process.

In Korea, the nationwide CRC screening was initiated as part of the National Cancer Screening Program (NCSP) in 2004 (Shim JI et al. Cancer Res Treat 2010;42:191-8). According to this program, all Koreans aged 50 years or older receive a single annual FIT as an initial CRC screening and a colonoscopy as a further examination if FIT is positive (Park MJ et al. Scand J Gastroenterol 2012;47:461-6). In the nationwide CRC screening, the FIT method itself has not changed since the beginning of NSCP, except that the proportion of quantitative test methods increased. As described in the method of our manuscript, FIT in this study included both quantitative and qualitative methods and there exists each cut-off level that determine positive for each product. Thus, it was expected the subjects with FIT between 2009 and 2011 did not have a significant effect on the results. We added this limitation in the Discussion section as follows: “Fourth, the subject's FITs period was old. It was difficult to use the most recent data because it took considerable time to organize and establish the insurance data in NHID and analysis of this data required several stages of approval and a waiting time.”

2.Your p-value threshold of 0.05 should be corrected to reflect the number of hypothesis tests conducted. Please ensure a correction is applied and justified within the manuscript, otherwise the conclusions could be misleading. 

Reply: Multiple comparisons were performed using the Benjamini and Hochberg methods for the results of each diagnostic performance indicator. The results were added in Table 3 and the following sentence was added in footnote: aThe false discovery rates (FDRs) were determined using the Benjamini and Hochberg test for multiple comparisons correction.”

The results for PPVs and sensitivities did not change even after multiple comparisons.

3.You specify combination treatment of antiplatelet agents and anticoagulants as an exclusion criterion. Would it not have been of interest to also investigate this group of patients?

Reply: Thank you for your comments. As recommended by the reviewer, we analyzed 5,539 patients who received combination treatment with antiplatelet agents and anticoagulants (warfarin). The results are shown in the table below. The PPV and sensitivity of FITs for CRC detection were not significantly different between “antiplatelet agents + anticoagulants users” and “matched non-users”. However, “antiplatelet + anticoagulant users” are a heterogeneous group that includes users of various types of antiplatelet agents. Our study sought to analyze in detail according to the type of antiplatelet agents, and thus, this group was not included in the analysis.

Drug types

The number of subjects

PPV

NPV

Sensitivity

Specificity

TP

FN

TN

FP

%

(95% CI)

RR

(95% CI)

P

%

(95% CI)

RR

(95% CI)

P

%

(95% CI)

RR

(95% CI)

P

%

(95% CI)

RR

(95% CI)

P

Users

Non-users

26 (0.47)

10 (0.18)

5,143 (92.85)

360 (6.50)

6.74 (4.24-9.24)

1.00 (ref)

99.81 (99.69-99.93)

1.00 (ref)

72.22 (57.59-86.85)

1.00 (ref)

93.46 (92.80-94.11)

1.00 (ref)

Users

19 (0.34)

10 (0.18)

5,063 (92.85)

447 (8.07)

4.08 (2.28-5.87)

0.63 (0.35-1.11)

0.110

99.80 (99.68-99.92)

1.00 (1.00-1.00)

0.978

65.52 (48.22-82.82)

0.89 (0.63-1.24)

0.492

91.89 (91.17-92.61)

0.98 (0.97-0.99)

0.002

4.The p-values presented in Table 2 are meaningless without any correction for multiple testing due to the large amount of comparisons undertaken. Indeed, the relevance of the information within this table is questionable given that the majority of non-drug-users were excluded from the analysis as they were removed during the matching process. My preference would be to exclude this table altogether, and just keep table 3 which relates to the actual participants studied. 

Reply: Thank you for this comment, we agree with your concern. As the reviewer recommended, we deleted Table 2. Instead, we presented the number, age, and sex of the entire study population without mentioning p values in Supplementary Table 1.

5.Did you consider any adjustments in your analyses for type of test undertaken, given that there was large variation in the type of test. 

Reply: Thanks for your thoughtful comments. In this regard, we reviewed the methodology and confirmed that analysis is necessary with the modified Poisson regression model mentioned in Zou's paper (Guangyong Zou, A Modified Poisson Regression Approach to Prospective Studies with Binary Data, American Journal of Epidemiology, 2004; 159 (7): 702–06). In this paper, it is described as follows; “When Poisson regression is applied to binomial data, the error for the estimated relative risk will be overestimated. However, this problem may be rectified by using a robust error variance procedure known as sandwich estimation, thus leading to a technique that I refer to as modified Poisson regression.” Therefore, the results were re-analyzed through the modified Poisson regression analysis, and the description was revised in the method section as follows: “Modified Poisson regression with robust error variance was then used to assess the differences in the performance of the FIT between drug users and non-users.”

6.Why did you choose Poisson regression modelling to test for difference in PPV between the two groups? In addition, what covariates were adjusted for in the regression model ? As you are comparing proportions it would be more usual to use a hypothesis test such as the chi-square test, or if wanting to adjust for covariates, a logistic regression approach. 

Reply: We applied a Poisson regression model to calculate relative risk by adjusting for sex and age based on incidence. When evaluating a binary dependent variable by controlling for the covariates, the analysis can be performed through logistic regression. However, if the rare event rate assumption is not met, the odds ratio calculated by the logistic model can differ significantly from the relative risk. We tested the PPV differences between the two groups (drug users and non-users), as well as a variety of other diagnostic indicators. Other diagnostic indicators except for the PPV are difficult to satisfy this condition. In addition, Zou's paper mentioned above explained that there is no need to rely on the logistic model when calculating the RR is a major concern. For these reasons, we used the Poisson regression model to generate the results.

7.What is the justification for undertaking an analysis of the overall dataset in addition to the matched analysis? The former seems to make the latter redundant. I would suggest that the overall analysis is excluded, and instead the paper focusses on the matched analysis only. 

Reply: As the reviewer recommended, we deleted the results for the entire study population (Tables 2, 5) and focused on the results for the 1:1 matched population.

8.What is the 'univariable' analysis referred to in Table 5 ? Is this the same as the 'crude' analysis ? Please ensure consistency of terminology throughout. Indeed, please ensure that all methods used in producing the reported results are reported in the methods section, for transparency. 

Reply: As mentioned in the above reply, we deleted Table 5 and related sentences. Accordingly, 'univariable' and 'crude' were not used throughout the manuscript.

9.In lines 316-317, on what basis are you claiming " almost all participants would have taken antithrombotic drugs on the day of as well as before the FIT. " As far as I am aware there were no measurements of extent of adherence available in the dataset and so this statement could be misleading. The authors may wish to explore known rates of adherence for the studied drugs, and comment accordingly on the potential impact of adherence on the study estimates. 

Reply: Thank you for this comment, we agree with your concern. We removed the misleading sentence from the manuscript.

10.Please provide a power calculation to illustrate available power for the various tests undertaken. 

Reply: We evaluated the power of various tests and confirmed sufficient power for the results that were significant. Thus, it was added the following description to the Result section. “In multiple comparison analysis, results with a P value of less than 0.05 met the general minimum requirements for power (more than 80%).”

Round 2

Reviewer 1 Report

Your reply it is good for me.